# LncRNAs in HCV Infection and HCV-Related Liver Disease

**DOI:** 10.3390/ijms21062255

**Published:** 2020-03-24

**Authors:** Juan P. Unfried, P. Fortes

**Affiliations:** Department of Gene Therapy and Hepatology, Center for Applied Medical Research (CIMA), University of Navarra (UNAV), E-31008 Pamplona, Spain; junfried@unav.es

**Keywords:** HCV, lncRNAs, type I IFN, HCC, liver fibrosis, biomarkers

## Abstract

Long non-coding RNAs (lncRNAs) are transcripts with poor coding capacity that may interact with proteins, DNA, or other RNAs to perform structural and regulatory functions. The lncRNA transcriptome changes significantly in most diseases, including cancer and viral infections. In this review, we summarize the functional implications of lncRNA-deregulation after infection with hepatitis C virus (HCV). HCV leads to chronic infection in many patients that may progress to liver cirrhosis and hepatocellular carcinoma (HCC). Most lncRNAs deregulated in infected cells that have been described function to potentiate or block the antiviral response and, therefore, they have a great impact on HCV viral replication. In addition, several lncRNAs upregulated by the infection contribute to viral release. Finally, many lncRNAs have been described as deregulated in HCV-related HCC that function to enhance cell survival, proliferation, and tumor progression by different mechanisms. Interestingly, some HCV-related HCC lncRNAs can be detected in bodily fluids, and there is great hope that they could be used as biomarkers to predict cancer initiation, progression, tumor burden, response to treatment, resistance to therapy, or tumor recurrence. Finally, there is high confidence that lncRNAs could also be used to improve the suboptimal long-term outcomes of current HCC treatment options.

## 1. Hepatitis C Virus (HCV)

HCV infection leads to the death of half a million people every year [1,2]. Although several antiretroviral agents have been recently developed that impede viral replication and lead to viral clearance in most patients, the high prevalence of the infection (around 2% of the world population is infected with the virus), the high number of undiagnosed patients and the slow progression to fatal symptoms, makes HCV infection a major gastrointestinal health problem [3]. After viral entry, HCV produces an acute infection that can be cleared spontaneously in some patients (15–50% of cases) while it may progress to a chronic infection in others (55–85%) [4]. Most chronically infected patients respond to infection with liver inflammation and liver injury that may cause liver fibrosis, cirrhosis, and, in some cases, hepatocellular carcinoma (HCC) [5]. These are the primary causes of death after HCV infection.

A careful description of the viral particle and the mechanisms that allow viral replication can be found in any of the several excellent reviews about HCV published recently (this Special Issue, [6]). Briefly, HCV belongs to the Flaviviridae family of positive, single-stranded, enveloped RNA viruses. The viral particle is small in size (40–80 nm) and circulates in the blood bound to lipoproteins and lipid particles, which can help viral evasion from the immune system and efficient infection of target hepatocytes (Figure 1) [7]. At the hepatocyte surface, HCV binds to different receptors, including LDLR (low-density lipoprotein receptor) and CLDN1 (Claudin-1), and is transported to the cytoplasm by endocytosis [8]. Once in the endosome, the low pH helps viral uncoating, and the viral genome is free to bind to the endoplasmic reticulum, where a membranous web is formed that helps viral replication [9]. Prior to replication, the viral genome is translated from a 5′ internal ribosome entry site (IRES) by cap-independent translation [10]. Translation produces a polyprotein that is cleaved into 3 structural proteins (core protein and E1 and E2 glycoproteins) and 7 non-structural proteins (p7, NS2, NS3, NS4A, NS4B, NS5A, and NS5B) that are required for polyprotein processing, viral replication, packaging and release, and for blocking the cellular antiviral response.

### HCV and the Antiviral Response

Efficient HCV replication requires that viral proteins block type I IFN (interferon) synthesis and signaling pathways (Figure 1). These are key routes of the type I IFN response, one of the primary innate immune weapons against microbe infection. IFNs have been classified in different groups according to the cellular receptor that they bind: type I (which includes IFNα and β), type II (IFNγ), and type III (IFNλ). However, there is a significant overlap in the genes that are induced after activation of the different IFNs [11]. A systematic description of the IFN response is away from the scope of this article and can be found in any of the outstanding reviews published recently [12,13]. In this review, we will describe the relationship between HCV and type I IFN.

The synthesis of type I IFN is the result of the activation of cellular sensors by PAMPs (pathogen-associated molecular patterns) such as DNA, RNA, or LPS (lipopolysaccharide). In the case of HCV, viral RNA is sensed soon after infection by the canonical sensor RIG-I (retinoic acid-inducible gene I, which binds to the viral genome) and by the non-canonical protein kinase R (PKR, which recognizes the viral 5′UTR) or by the DEAD-box helicase 3 X-linked (DDX3X, activated by the viral 3′UTR) (Figure 1) [14,15,16,17]. RIG-I and the canonical sensor TLR3 (toll-like receptor 3) can also be activated by the viral dsRNA (double-stranded RNA) produced during replication [18,19]. In addition, it has been recently shown that the canonical sensor MDA5 (melanoma differentiation-associated protein 5) can also bind viral RNA and activate IFN [20]. Activated sensors can bind MAVS (mitochondrial antiviral signaling protein) and TRIF (Toll-Interleukin Receptor-domain-containing adapter-inducing interferon-β), which trigger the induction of transcription factors NF-κB (nuclear factor κb) and IRF (interferon regulatory factors), in charge of activating IFN synthesis. 

IFN signaling induces the expression of numerous genes with potent antiviral functions at different levels (Figure 1). Secreted type I IFN binds to the IFNAR (IFN-α/β receptor), activating JAK/STAT (Janus kinase/signal transducers and activators of transcription) and phosphorylation, dimerization, and transport to the nucleus of STAT1 and STAT2 (signal transducer and activator of transcription 1 and 2). There, they are bound by IRF9 to constitute the ISGF3 (IFN-stimulated gene factor 3) complex, in charge of activating the transcription of the vast ISG (IFN-stimulated genes) repertoire [21,22]. Several ISGs have been shown to decrease HCV replication by affecting virus infection (Mx, TRIM, IFITM, CH25H), viral RNA translation, replication, stability (OAS, IFIT, GBP1), or virus packaging and release (tetherin/BST2, viperin) [23,24]. These effects are reinforced by ISGs that contribute to IFN synthesis or signaling. Indeed, expression of PKR, STAT1, STAT2, or IRFs is also induced by IFN, resulting in a positive loop that amplifies IFN response. Interestingly, several ISGs inhibit the IFN response to allow the cell to return to homeostasis [25,26]. One of them is SOCS3 (suppressor of cytokine signaling 3) that blocks JAK activity and STAT binding. Therefore, these inhibitory ISGs can be considered as true proviral factors induced by IFN.

Activation of some of these inhibitory ISGs is one of the mechanisms employed by HCV to counteract the IFN pathway. Viral core protein induces SOCS3 and PP2A (Protein Phosphatase 2), which also blocks STAT1 function [27,28,29]. In addition, several HCV proteins have evolved to block the function of specific antiviral ISGs or to block IFN synthesis: MAVS and TRIF are cleaved by the viral NS3-NS4A protease, and RIG-I pathway is blocked by HCV-mediated induction of autophagy [30,31,32]. Despite this and against initial expectations, many patients with active HCV infections have high levels of ISG mRNAs [33]. One possibility to explain this observation is that ISG mRNAs are not efficiently translated in HCV-infected cells due to PKR activation. Although PKR activation by the viral genome may seem damaging for the HCV cell cycle, it is an excellent weapon to increase viral replication. PKR activates the IFN synthesis pathway but also phosphorylates eIF2α (eukaryotic translation initiation factor 2 alpha) leading to inhibition of cap-dependent translation. As viral translation is cap-independent, it is not affected by PKR action [34]. Instead, translation of newly transcribed ISG mRNAs will be abrogated. Similar to PKR, two other ISGs with a proviral function in HCV infection are ISG15 and DDX3X, whose activation results in increased levels of lipogenic genes required for viral packaging [17,35,36]. ISG15 is particularly interesting. ISG15 is required for protein ISGylation, an IFN-induced ubiquitin-like process that serves to modify newly-translated proteins co-translationally [35]. After infection, high levels of viral proteins need to be translated, and viral protein ISGylation should alter viral protein stability or structure, and its function [37]. However, ISG translation is also mandatory in newly infected cells. In fact, RIG-I ISGylation affects functionality and leads to reduced IFN response [35,38]. Therefore, the battle between viral replication and the antiviral response causes severe collateral damage over cellular and viral proteins, as their stability and function are compromised by ISGylation, and their translation is blocked by PKR. Under such circumstances, it is plausible that cellular and viral evolution has fostered the development of non-coding RNAs that are resistant to such protein-hostile environments and could function to modulate viral replication and the antiviral response [39]. 

## 2. Long Non-Coding RNAs (lncRNAs)

Wrongly depicted as “junk DNA”, the non-coding genome corresponds to the fraction of a genome that does not code for proteins, which in humans covers around 98% of our genomic sequences. Despite its non-coding nature, up to 90% of human non-coding sequences are transcribed into RNA molecules that have been termed together as non-coding RNAs (ncRNAs) [40]. These RNAs have been assigned to different categories according to biological and functional features. Some examples are ribosomal RNAs (rRNAs) and transfer RNAs (tRNAs), involved in protein synthesis, small nuclear RNAs (snRNAs), part of the splicing machinery, and small nucleolar RNAs (snoRNAs), which assist in the modification of other ncRNAs [41]. Additionally, ranging from 20 to 30 nucleotides in length, microRNAs (miRNAs), small interfering RNAs (siRNAs), and PIWI-interacting RNAs (piRNAs) can base-pair with complementary RNA sequences and mediate transcriptional and post-transcriptional silencing [41,42]. Several miRNAs have been shown to play essential roles in HCV replication and pathogenesis. More information about this can be found in the excellent reviews about this topic published recently ([24,43,44], and this Special Issue). In recent years, a new species of longer ncRNAs have emerged. They are collectively called long non-coding RNAs and, although still poorly understood, they are the most numerous and functionally diverse family of ncRNAs [45,46].

As their name indicates, lncRNAs are long (200 nucleotides or more) and show no significant protein-coding capacity. Despite operatively belonging to the family of ncRNAs, many lncRNAs are biochemically indistinguishable from messenger RNAs (mRNAs), their protein-coding counterparts. In fact, most of what is known about lncRNA biology comes from comparative studies between lncRNAs and mRNAs. In general, lncRNAs and mRNAs are both transcribed by RNA polymerase II from loci with similar chromatin marks, and they are often 5′-capped, spliced, and polyadenylated [47]. Except for the presence of translated open reading frames (ORFs), only general trends separate lncRNAs from mRNAs. For example, lncRNAs tend to be shorter and have fewer but longer exons. Compared to mRNAs, lncRNAs are less conserved, their expression is enriched in the nucleus, and they are, overall, expressed to lower levels and in a more tissue-specific manner [48,49]. 

LncRNAs are challenging to classify due to the diversity of their biogenesis and functionality. Historically, they have been classified according to their genomic position and their mechanism of action. Taking into account their genomic organization, lncRNAs are classified in reference to their nearest protein-coding gene or their proximity to promoters and enhancers, generating a broad definition of lncRNA biotypes (Figure 2) [50]. In this context, lncRNAs can be found in isolated regions of the genome (lincRNAs) or linked to DNA regulatory features such as enhancers (eRNAs), promoters (PROMPTs), or coding genes. In the latter case, lncRNAs can be transcribed in the same (sense) or the opposite (antisense) orientation in reference to the coding gene. In addition, antisense lncRNAs (NATs) can be convergent, divergent, or bidirectional, and they can be linear or circular and may be very long (vlincRNA). 

LncRNAs function through a multitude of mechanisms, which can be broadly classified into those that regulate local chromatin structure and gene expression in *cis*, versus those that leave the site of transcription to perform cellular functions in *trans* [51]. At least three potential mechanisms can be described for a lncRNA locus that locally regulates chromatin or gene expression: (1) the lncRNA transcript itself regulates the expression of neighboring genes through the recruitment of regulatory factors to the locus, (2) the process of transcription or splicing of the lncRNA confers a regulatory function that is independent of the sequence of the RNA transcript, or (3) regulation in *cis* depends solely on DNA elements within the lncRNA promoter or gene locus and is entirely independent of the encoded RNA or its production [51]. On the other hand, a high number of trans-acting lncRNAs are thought to exist. They leave the site of transcription and operate elsewhere in the cell through three basic mechanisms: (1) by regulating chromatin states and gene expression at regions distant from their transcription site, (2) by influencing nuclear structure and organization, and (3) by interacting with and regulating the behavior of proteins and/or other RNA molecules [51].

Notably, lncRNAs perform their regulatory functions in an exceptionally cell type- and cell state-specific manner. Transcriptomic analyses have allowed the discovery of subsets of lncRNAs whose expression is highly restricted to a cell or tissue type but also to discreet developmental or disease stages, suggesting a tighter transcriptional regulation compared with protein-coding genes [52,53,54]. This serves as evidence to support an essential role for lncRNAs in determining and maintaining cell states, and this has been particularly observed in cancer [55]. In fact, several lncRNAs such as HOTAIR (homebox C transcript antisense RNA) or PVT1 (lncRNA plasmacytoma variant translocation 1) behave as oncogenes [56,57], whereas others such as MEG3 (maternally expressed 3) or PANDAR (promoter of CDKN1A antisense DNA damage-activated RNA) act as tumor suppressors [58,59]. Interestingly, it has been shown that most lncRNAs deregulated in cancer are tumor-specific, whereas only a handful behave as “onco-lncRNAs”, deregulated across many cancer types [54,60,61]. This has encouraged the development of a whole new field for the exploration of lncRNAs in cancer as a source of novel molecular targets and as biomarkers for diagnosis [62], predictors of treatment response [63], disease-free and overall survival [64,65] and risk of metastasis [66]. Similarly, deregulation and functional implications of lncRNAs have been studied in many other diseases, including HCV infection. 

## 3. General Considerations of HCV and lncRNAs

Hepatocytes are the primary targets of the HCV virus. Most HCV studies are performed with liver samples from infected patients or using cultured liver cells, which allow viral entry, replication, and release. Until the complexity of a human liver can be recapitulated in the laboratory, it is unclear whether tissue cultures faithfully reflect liver infection. For the time being, it is reasonable to study HCV effect on liver fibrosis, cirrhosis, and HCC using patient samples, as these diseases involve hepatocytes and surrounding cells, and are challenging to recreate in culture. Instead, the HCV cell cycle can be studied in cultured cells, and the results obtained should be validated with patient samples. 

The lncRNA transcriptome is highly deregulated after HCV infection [67]. This is caused by the expression of viral lncRNAs and the deregulation of cellular RNAs in response to viral replication or to the antiviral response.

## 4. HCV lncRNAs

Further studies should be carried out to determine whether the HCV genome can be processed in several ways to produce viral lncRNAs. Preliminary analyses do not support the presence of viral genomes with internal deletions or fusions between cellular and viral RNAs. Instead, the 5′ end of the IRES sequence can be processed by the cellular endoribonuclease XRN1 (5′-3′ exoribonuclease 1) [68]. This results in subgenomic viral RNAs that cannot be translated and therefore are bona fide viral lncRNAs of unknown function.

## 5. Cellular lncRNAs Related to HCV Cell Cycle and the Antiviral Response 

### 5.1. LncRNAs that Affect the Viral Cell Cycle 

Few lncRNAs have been described that increase after HCV infection and reduce viral replication (Figure 1, Table 1). GAS5 (growth arrest-specific transcript 5) is an excellent example. In human embryonic stem cells, GAS5 promotes self-renewal [69], whereas in most cells, GAS5 regulates growth arrest and apoptosis by binding to the activated glucocorticoid receptor and impeding its function as a transcription factor [70]. GAS5 is also a decoy in HCV infection. GAS5 is upregulated after infection, and GAS5 5′ end binds viral protein NS3 and blocks NS3 function [71]. NS3 belongs to the replicase complex. Its helicase activity could unwind viral RNA for replication and viral assembly [72]. In addition, NS3 is a protease involved in processing the non-structural proteins of HCV and cleaving cellular factors that activate the immune response, such as MAVS and TRIF [73]. GAS5 induction and NS3 blocking lead to increased levels of MAVS and TRIF and to increased ISG expression. Therefore, GAS5 plays an important antiviral role in HCV infection.

Most lncRNAs upregulated in HCV infection described to play a role in the viral cycle are proviral factors that contribute to viral release (Figure 1, Table 1). One of them is HOTAIR, a well-studied lncRNA involved in cancer (see below). HOTAIR regulates chromatin and gene expression in *trans* by binding to PRC2 (polycomb repressive complex 2) [57]. In HCV-infected cells, HOTAIR expression is induced by the core protein and functions to silence SIRT1 (sirtuin 1) promoter, alter lipid metabolism, and contribute to efficient viral release [74]. HULC (highly upregulated in liver cancer) is also a lncRNA upregulated in several tumors (see below) that facilitates lipid biogenesis and viral release [75]. HULC upregulation after HCV infection results from RXRA transcription (retinoid X receptor alpha). Similarly, other lncRNAs are upregulated by transcription factors induced by infection. This is the case of STAT3 and lncRNAs IGF2-AS (insulin-like growth factor 2 antisense RNA) and 7SK (RNA component of 7SK nuclear ribonucleoprotein), which also promote viral release by increasing the levels of PI4P (phosphatidylinositol 4-phosphate) [76,84].

### 5.2. LncRNAs that Affect the Antiviral Response

Most of the lncRNAs described to date that are deregulated in cultured infected cells and that have been functionally studied play a role in the antiviral response (Figure 1, Table 1). This highlights the relevance of regulating this pathway to allow or to block efficient viral replication. In addition, this reinforces the notion that lncRNAs could play a prominent role as positive or negative regulators of the cellular antiviral pathway [39]. Preliminarily, it is reasonable to think that lncRNAs induced by viral replication should play proviral functions, whereas lncRNAs induced by the IFN synthesis or signaling pathways should function as antiviral molecules. However, this is not always the case. LncRNAs induced by IFN may also act as negative regulators of the pathway that help the cell return to homeostasis. These IFN-induced lncRNAs may function as proviral factors. Moreover, in some instances, it is difficult to discriminate whether lncRNAs have been induced by viral replication or by the PAMPs produced during replication. As a general rule, lncRNAs whose expression is altered by the antiviral pathways should show deregulation in response to infection with several other viruses, and/or in response to treatment with IFN and/or different PAMPs. This is the case of lncRNA TSPOAP1-AS1 (TSPOAP1 antisense RNA 1) and MIR155HG (miRNA-155 host gene), upregulated in response to infection with influenza and other viruses and by the dsRNA mimic pI:C (lncRNA TSPOAP1-AS1) [85,86]. Interestingly, lncRNA TSPOAP1-AS1 inhibits IFNβ and ISG transcription, playing a proviral role, whereas MIR155HG functions as an antiviral by inhibiting PTP1B (protein tyrosine phosphatase 1B), resulting in higher production of IFNβ and several ISGs.

#### 5.2.1. IFN-Related lncRNAs

Although their effect on HCV replication has not been directly addressed, several lncRNAs have been described as being induced by IFN signaling or synthesis pathways. This should have an impact on the HCV cell cycle as they affect the antiviral response. In this review, we highlight those whose functional mechanism has been studied. They target RIG-I, IRF3, and IFNβ transcription. lncLRRC55-AS (lncRNA antisense to leucine rich repeat containing 55) is upregulated by IFN treatment and by infection with several viruses in mice [87]. Interestingly, lncLRRC55-AS binds PME-1/CSTB (phosphatase methylesterase 1/cystatin B) and helps for PME-1-mediated demethylation and inactivation of PP2A. PP2A is a phosphatase in charge of IRF3 dephosphorylation and inactivation. Therefore, lncLRRC55-AS leads to IRF3-mediated transcription and ISG expression and acts as an antiviral factor. NEAT1 (nuclear paraspeckle assembly transcript 1) can also function as an antiviral lncRNA, as it is induced by RIG-I and IRF7 and helps to increase RIG-I and DDX60 (DExD/H-Box helicase 60) expression and IFN production [88]. In turn, lnc-Lsm3b (lncRNA U6 snRNA-associated Sm-like protein) is an IFN-induced lncRNA that blocks the shift in the conformation of RIG-I, required for signaling, limiting IFNβ production [89]. Acting downstream, lnc-MxA (lncRNA Interferon-induced GTP-binding protein MxA) is also an IFN-induced lncRNA that interferes with IFNβ transcription by forming a DNA-RNA triplex helix at the IFNβ promoter [90]. Therefore, lnc-Lsm3b and lnc-MxA are IFN-induced lncRNAs that help viral replication. 

#### 5.2.2. IFN-Related lncRNAs that Affect HCV Replication

Other IFN-induced lncRNAs have been studied for their effect on HCV-replication. They can be divided into three major classes: (i) lncRNAs affecting the IFN synthesis pathway, (ii) lncRNAs affecting the expression of specific ISGs located nearby, and (iii) lncRNAs that act on ISG transcription in a more general manner.

#### 5.2.3. LncRNAs Affecting the IFN Synthesis Pathway

Similar to what has been previously described (see above), lncRNAs induced by IFN can affect the IFN synthesis pathway to increase or decrease HCV replication (Figure 1, Table 1). lncATV/AL391832.2 binds to RIG-I and is a potent negative regulator of RIG-I signaling, leading to decreased IFNβ and antiviral factors and increased HCV replication [77]. In turn, lncITPRIP-1, (Inositol 1,4,5-Trisphosphate Receptor Interacting Protein) annotated as CFAP58-DT (CFAP58 divergent transcript) blocks HCV replication by binding to the carboxy-terminal domain of MDA5, promoting MDA5 oligomerization and activation, and increasing ISG levels [20]. Interestingly, it has been recently shown that MDA5 sensor has a potent antiviral function against HCV by a dual mechanism. On the one hand, MDA5 recognizes viral RNA and induces the cascade that leads to IFNβ synthesis. On the other hand, MDA5 binds to viral RNAs and interferes with RNA replication [20]. CFAP58-DT strengthens the binding of MDA5 to viral RNA and contributes to the two mechanisms that mediate MDA5 antiviral function. 

#### 5.2.4. LncRNAs Affecting the Expression of Specific ISGs Located Nearby

Several lncRNAs have been described as acting by regulating the expression of neighboring genes, including several induced by IFN (Figure 1, Table 1) [79,91]. This is the case of two lncRNAs that affect HCV replication. AL445490.1 is induced by IFN and is antisense to intron 1 of IFI6 (interferon alpha inducible protein 6) [78]. In fact, AL445490.1/lncRNA-IFI6 is a nuclear lncRNA that induces histone modification at IFI6 promoter to hinder transcription. IFI6 is a well-known ISG that plays a role in apoptosis, and that affects the epidermal growth factor receptor (EGFR)-mediated interaction of CD81 with the HCV receptor CLDN1 [92]. Therefore, IFN signaling leads to lncRNA-IFI6 expression, which blocks IFI6 transcription and favors HCV entry. Surprisingly, lncRNA-IFI6 can also decrease the expression of IFI6 when it is overexpressed from a plasmid, suggesting that it is not a bona fide *cis*-acting lncRNA and that a mechanism should exist to guide this lncRNA to IFI6 promoter.

Similar results have been obtained with BISPR (BST2 IFN-stimulated positive regulator). BISPR is a lncRNA induced by IFN [79,80,81] expressed from a bidirectional promoter that also drives BST2/tetherin. After IFN signaling, BISPR is expressed first, accumulates in the nucleus, and can act in *trans* to activate transcription of *BST2*, probably, by counteracting the suppressive role of PRC2 [80]. BST2/tetherin is a well-known ISG that impedes the budding of viral particles by attaching virions to the cell surface and allowing their degradation by the lysosomes [93,94]. Therefore, BISPR action on *BST2* should abrogate viral release. 

#### 5.2.5. LncRNAs Affecting General ISG Transcription

Several examples exist of IFN-induced lncRNAs that affect ISG transcription directly (Figure 1, Table 1). Most of those described work to decrease the levels of ISGs, with lncRNA-32/LUARIS (lncRNA upregulator of antiviral response interferon signaling) being the exception [81]. LUARIS is highly induced by pI:C and IRF3 activation but, surprisingly, inhibited with IFNβ treatment. After transcription, hnRNPU (heterogeneous nuclear ribonucleoprotein U) binds to and stabilizes LUARIS. In addition, LUARIS interacts with ATF2 (activating transcription factor 2), a transcription factor that regulates the expression of several genes, including those involved in anti-apoptosis, DNA damage response, cell growth, and inflammation [95,96]. Binding to LUARIS is required for ATF2 interaction to a cAMP-responsive element located in the intron of IRF7 and IFN-independent ISG expression. Thus, in the absence of type I IFN, pI:C sensing can activate ISG expression through LUARIS activation and ATF2 activity. This has a strong negative impact on the replication of hepatitis B virus (HBV) and HCV. Indeed, Viperin is among the ISGs profoundly affected by LUARIS. Viperin is a strong antiviral protein for HCV, as it binds to NS5A in the replication complex, blocking NS5A function in viral replication and viral packaging [97,98].

Viperin is also highly affected by a lncRNA located nearby called lncRNA-CMPK2/NRIR (negative regulator of the IFN response) [82]. NRIR induction by IFNα leads to decreased levels of the expression of neighboring genes *Viperin* and *CMPK2*, but also of other ISGs located away from NRIR locus, such as *ISG15*, *IFIT3*, *CXCL10*, and *IFITM1*. Interestingly, this is also observed in the absence of NRIR induction by IFNα. Therefore, NRIR expression leads to nuclear accumulation and transcription initiation inhibition of a specific set of ISGs, not all, acting as a negative regulator of the antiviral response. As expected, NRIR levels are increased in the liver of patients infected with HCV compared to controls. In infected cells, NRIR acts to decrease antiviral factors, such as viperin, contributing to viral replication.

A similar role in ISG regulation has been shown for lnc-ITM2C-1/lncR8/GCSIR (Integral membrane protein 2C/G protein-coupled receptor 55 (GPR55) *cis* regulatory suppressor of immune response RNA) and EGOT (eosinophil granule ontogeny transcript). GCSIR is upregulated after HCV infection and downregulated by type I IFN and contributes to viral replication by activating the expression of the neighboring gene *GPR55* [83]. GPR55 is a cannabinoid-sensitive receptor with a significant role in the central nervous system. There, the endocannabinoid (EC) system is involved in the immune response that follows injury. In addition, several studies have linked cannabis consumption and the EC system with liver disease [99]. EC system plays a role in inflammation, steatosis, and liver fibrosis, and increases in patients with liver disease [100]. In HCV-infected patients, the levels of ECs increase in plasma, and the EC system is associated with fibrosis progression and with the immune response. Indeed, in cultured cells, GPR55 represses expression of ISGs, including *IFITM1*, *ISG15*, and *Mx1*. In summary, the induction of GCSIR by HCV infection leads to increased expression of GPR55, downregulation of ISGs, and increase viral replication [83].

EGOT is a lncRNA whose sequence and structure have been conserved during evolution, and that has been studied in several settings [101]. In mature eosinophils, EGOT regulates the levels of toxic proteins [102]. In cancer, EGOT is downregulated in kidney tumors, where high levels correlate with better survival [103]. In HCV-infected cultured cells and patients, EGOT is highly upregulated. EGOT is also upregulated in cells infected by other RNA viruses such as influenza or SFV (Semliki Forest virus), but not DNA viruses such as adenovirus of HBV ([67] and Carnero et al., unpublished). Indeed, viral RNA is sensed by PKR and RIG-I that activate NF-κB, which is a major driver of EGOT expression. Similar to previously described for NRIR and GCSIR, EGOT expression in infected cells leads to decreased levels of several ISGs and increased viral replication [67]. Similar results are observed with cells infected with SFV, another RNA virus. Surprisingly, the opposite result is obtained in cells where the IFN signaling pathway is highly induced (Barriocanal et al., submitted). After activation with pI:C, EGOT levels increase, and NF-κB transcription is activated. In these settings, EGOT contributes to NF-κB transcription, and, as the IFN pathway is induced, NF-κB synergizes with STAT1/STAT2 and IRF9, leading to higher levels of ISGs. In turn, after HCV infection, the IFN pathway is inhibited, and EGOT functions to decrease ISG expression. Therefore, simply by modulating the IFN response, HCV has evolved to change the cellular antiviral function of EGOT into a proviral function. 

In summary, most of the lncRNAs deregulated after HCV infection function to increase HCV replication by blocking the IFN antiviral response at different levels (Figure 1, Table 1). In addition, given that many of these lncRNAs are induced by the IFN pathway, it is conceivable that they could function as negative regulators of its own expression, resulting in a negative regulatory loop that helps IFN-treated cells return to homeostasis. 

## 6. LncRNAs in HCV-Related Liver Diseases

### 6.1. LncRNAs Related to Extrahepatic Manifestations and HCV-Related Liver Fibrosis

HCV infection is primarily a liver disease. However, several extrahepatic manifestations have been associated with chronic HCV infection (CHC), including many autoimmune diseases [104]. The role of lncRNAs in HCV-infected patients has overlooked extrahepatic manifestations despite evidence pointing to the contrary. A good example is lncRNA-AF085935, which has been recently described to have potential as an early marker of rheumatoid arthritis [105]. In the liver, few studies have focused on lncRNAs related to HCV-induced fibrosis. One of them has described lncRNA-ATB (long noncoding RNA activated by TGFβ (transforming growth factor-beta), a TGFβ-induced well-known oncogenic factor, as highly upregulated in the liver and blood samples of patients with HCV-related liver fibrosis [106,107]. In these patients, plasma levels of lncRNA-ATB correlated significantly with liver fibrosis stage. Interestingly, increased levels of lncRNA-ATB were also observed in activated hepatic stellate cells, which produce the extracellular collagen deposits that accumulate in fibrotic scars. Furthermore, inhibition of lncRNA-ATB leads to decreased levels of COL1A1 (alpha-1 type I collagen) and of the stellate cell activation marker α-SMA (alpha-smooth muscle actin), probably by downregulating TGFβ signaling in these cells.

### 6.2. LncRNAs Related to HCV-Related Hepatocellular Carcinoma (HCV-HCC)

Most of the lncRNA research in HCV-infected patients has focused on HCV-HCC. Certainly, this makes sense, as malignant transformation is an exceptional inducer of lncRNAs [45], and HCV remains a significant risk factor for HCC development (Figure 3) [108]. As many as 20% of HCV-infected patients can develop cirrhosis over a 20 year period with a risk of up to 7% annual progression to HCC [109,110]. Moreover, incidence and mortality due to HCC have been increasing for the last 25 years. By 2030, it is predicted that the barrier of 1 million deaths per year will be surpassed, consolidating liver cancer as the fastest-rising cause of cancer-related deaths worldwide [111,112]. Patients with HCC are diagnosed at different stages, frequently presenting comorbidities that impoverish prognosis and complicate therapy allocation. HCC can be treated successfully in only 30%-40% of the cases that are detected at early stages of the disease [113]. In this setting, treatment options include resection, ablation, and liver transplantation. Unfortunately, tumor recurrence is high, and most HCCs are diagnosed at later stages when only chemoembolization and systemic therapies are available, which only increases patient survival for a few months [113].

Many transcriptome analyses have documented the outstanding aberrant expression of lncRNAs that accompany immunomodulatory dysregulation, altered DNA damage response, and cell metabolism disorders in HCC cancer cells [54,114]. In fact, expression profiling can unambiguously segregate tumors from healthy tissue by unsupervised clustering, suggesting a radical difference in the global non-coding transcriptome of HCC [115,116]. Similar studies have allowed significant progress in characterizing various functions of lncRNAs in the hallmarks of HCC development and progression [117,118]. However, substantial research is still needed in this field. Despite encouraging advances, the functions and mechanisms of most lncRNAs deregulated in HCC remain to be investigated, as there is hope that they will aid in a much-needed alternative outlook to HCC diagnosis and treatment. In this regard, lncRNA factors regulated by HCV that contribute to HCC development remain widely unexplored and are summarized in this review. For lncRNAs related to HCC of different etiologies, we recommend to the reader some of the excellent reviews published recently in this field [116]. 

Only a handful of studies have addressed the role of lncRNAs in HCV-related HCC (Figure 3, Table 2). Admittedly, such a study requires an outstanding collection of samples that need to include matching peritumoral tissue and tumor samples of uninfected patients and patients with reliable indicators of HCV infection such as the presence of viral RNA by PCR and/or HCV seropositivity. The first of such studies used microarray data to investigate lncRNAs with deregulated expression in HCV-infected patients with preneoplasic lesions (cirrhosis and hepatocellular dysplasia) and early or advanced HCC [119]. This study identified seven lncRNAs deregulated between preneoplasic samples and tumors. One of them, LINC01419, was later validated to be significantly upregulated in both HBV- and HCV-related HCC patient samples. Interestingly, LINC01419 was linked to cell cycle regulation by co-expression network analysis. This has been validated recently [120]. Further, LINC01419, renamed as PRLH1 (p53-regulated lncRNA for homologous recombination (HR) repair 1), was found to promote cell proliferation in p53-mutated HCC cells by acting as a platform for the HR machinery of DNA damage repair, favoring progression in the cell cycle. Comparison of samples from patients with early and late stages of HCC allowed the identification of lncRNAs AK021443 and AF070632, significantly up- and downregulated, respectively, between early and advanced HCC. This suggests that lncRNAs can be found deregulated in HCV-HCC hepatocarcinogenesis but also in the progression of the disease. In fact, AK021443 upregulation was later found to be associated with clinicopathological parameters and to be predictive of worse prognosis in HCC patients [121]. AF070632 did not validate in patients, but this could be attributed to its specific disease stage downregulation, which was not accounted for in the tumor biopsies. 

Zheng et al. [123] used TCGA data to identify deregulated lncRNAs in HCC samples from patients infected with HBV, HCV, or both. Remarkably, in all cases, a collection of 10 lncRNAs was significantly deregulated. However, no specific association was observed for any of the viral agents, suggesting that these lncRNAs are more hepatitis-specific than virus-specific. Only lnc-AFM-2/LINC02449 has since been further characterized in HCC. LINC02449 has potential diagnostic and prognostic value and inhibits HCC cell proliferation, migration, and invasion [124].

Finally, in a targeted approach, Zhang et al. [117] profiled 101 disease-related lncRNAs by qRT-PCR in patients infected with HBV, HCV, or HDV (hepatitis delta virus) using non-HCC cirrhotic liver biopsies as controls. Six lncRNAs were deregulated twofold in HCV-HCC. However, only two lncRNAs were found exclusively downregulated in HCV-HCC. One of them is aHIF (lncRNA antisense to HIF1A (hypoxia-inducible factor 1 alpha)). aHIF or HIF1A-AS has been described as being deregulated in several other cancer types, but its function in HCC is still unknown. The second candidate, PAR5/PWAR5 (Prader Willi/Angelman region RNA 5), has also been associated with other carcinomas, and acts by reducing EZH2 activity, although its potential role in HCC awaits validation. The lncRNA TMEVPG1 (Theiler’s murine encephalomyelitis virus persistence candidate gene 1) recently annotated as IFNG-AS1 (IFNG antisense RNA 1) was found to be downregulated in both HCV- and HDV-related HCC. This is in accordance with recent findings since IFNG-AS1 is transcribed antisense to *IFNG,* activating IFN-γ production, and the cellular response to viral infections by working as a chromatin structure modifier [122,139]. The three remaining lncRNAs, BC017743, BC043430, and LINC01152, are common to all viral etiologies. Only LINC01152 has been further described in another study as upregulated in HBV-infected patients and in cultured cells expressing HBV-encoded protein x (HBx) [140]. In this case, LINC01152 overexpression increased HCC cell proliferation and tumor formation in nude mice, probably by activating the transcription of *IL-23* (interleukin 23), leading to increased activation of the STAT3 pathway. Surprisingly, LINC01152 was described by Zhang and colleagues to be downregulated in all viral hepatitis-related HCC through an unknown mechanism. 

This discrepancy, the fact that LINC01152 is found upregulated in one study and downregulated in other, could be attributed to many factors that are common to lncRNA research: the high heterogeneity observed in HCC; the high specificity of lncRNAs that depends on etiology as much as on other epidemiological and environmental factors; or the inadequate resection, handling, or classification of patient samples. Further, it is intriguing how little overlap has been found among the studies evaluated thus far. In this case, it should also be considered that lncRNAs constitute the main output of the human genome. Current annotations are around 100,000 lncRNA loci according to NONCODE (v5) [141], many await annotation, and more are predicted to exist. 

Other approaches have decided to evaluate well-characterized lncRNAs in the context of HCV-related HCC (Figure 3, Table 2). Such is the case of HULC and MALAT1 (metastasis-associated lung adenocarcinoma transcript 1), which have been recently interrogated for their prognostic value [125]. Although both are significantly upregulated in HCC, stratification by expression levels in CHC patients was able to significantly correlate with recurrence-free and overall survival only for HULC and not for MALAT1. Interestingly, when the levels of MALAT1 were evaluated in serum samples, a significant correlation with clinicopathological features was validated in HCV-HCC patients. In this study, the authors propose that circulating MALAT1 is a putative non-invasive prognostic biomarker of liver failure [127].

### 6.3. LncRNAs in Liquid Biopsies as Biomarkers in HCV-HCC 

Like MALAT1, other lncRNAs are emerging as promising circulating biomarkers for diagnosis, prognosis, and response to therapy in cancer [142]. Current diagnostic and follow-up strategies for HCC rely heavily on medical image examination and AFP (alpha-fetoprotein) serum levels. These techniques are constrained in their sensitivity, specificity, and early detection prowess. Therefore, the gold standard for HCC confirmation remains the histopathological analysis of tissue biopsy samples. Tissue biopsy extraction is an invasive procedure with significant risks associated; therefore, they are limited by sampling frequency and size, causing incomplete representation of the tumor bulk [143]. On the contrary, liquid biopsies are minimally invasive because they allow the analysis of tumor components (circulating tumor cells, DNA, RNA, and extracellular vesicles (EVs) such as exosomes) that are released to body fluids such as the blood. Accumulating evidence supports that liquid biopsies allow the tracking of the evolutionary dynamics and heterogeneity of tumors, and they can also be used to track the development of resistance to therapy, the presence of residual disease, or its recurrence [144]. The clinical utility of liquid biopsies still needs to be fully validated. This process can be accelerated by the validation of new analytes such as non-coding RNAs that can be used in combination with other markers to improve its clinical utility but also their analytical strength [145,146]. This is encouraging for the field of lncRNAs because several studies have concluded that the lack of prognostic value from tissue sample validation is not necessarily predictive of the circulating biomarker potential [127]. 

Although lncRNAs have already been studied as biomarkers in HCC (reviewed in [142]), only a handful have shown biomarker potential for HCV-HCC (Figure 3, Table 2). One of them is UCA1. Urothelial carcinoma associated-1 or UCA1 has been validated to be an independent predictor of HCV-HCC risk and its serum expression levels have prognostic value for advanced clinical parameters of HCV-HCC [62]. Remarkably, the sensitivity of UCA1 to segregate HCV-HCC patients from HCV-infected patients was dramatically improved after considering the levels of AFP and lncRNA WRAP53 (Trp-Asp (WD) repeat containing antisense to TP53 (tumor protein p53)). Furthermore, lncRNA WRAP53 is an independent predictor of recurrence-free survival. 

To evaluate the loading of lncRNAs into exosomes and its predictive value, Zhang et al. [131] analyzed the levels of HEIH (HCC up-regulated EZH2-associated long non-coding RNA) in serum and exosomes from CHC, HCV-related cirrhosis, and HCV-HCC. Interestingly, although the expression of HEIH was significantly increased in serum and exosomes of HCV-HCC patients, the ratio of expression in serum versus exosomes was decreased in HCC compared to CHC, suggesting that the compartment of lncRNA accumulation can impact their prognostic value.

Similarly, other studies have analyzed TUG1 (taurine up-regulated 1) and CASC2 (cancer susceptibility 2) lncRNAs and have found that they correlate with disease stage and serum AFP. Importantly, TUG1 and CASC2, up- and downregulated in HCC, respectively, are known to help HCC cell proliferation and tumorigenicity in vitro [135]. Recently, classic HCC lncRNAs HOTAIR and HOTTIP (Homeobox A distal transcript antisense RNA) have also been jointly analyzed as biomarkers for HCV genotype 4-induced HCC with rather low sensitivity and specificity of around 60% and 85%, respectively, stimulating further validation in larger cohorts. This is encouraged by the fact that cell-based studies strongly suggest a relevant implication of both lncRNAs in HCV-HCC.

### 6.4. LncRNAs with Potential HCV-HCC Roles From Cell Studies

Although the biomarker potential of HOTAIR and HOTTIP falls somewhat short, evidence from in vitro studies supports a direct or indirect relationship with HCV. HOTTIP has been involved in a regulatory axis with miR-204 upon HCV genotype 1b core protein overexpression in HCC cells [147]. In addition, HOTAIR, upregulation in HCC, has been recently implicated in the epigenetic repression of miR-122 by DNA methyltransferase-mediated methylation of its promoter [137]. miR-122 is a liver-specific regulator of liver development whose repression is associated with HCV infection [148], HCC [149], and HCC chemoresistance [150]. Therefore, this finding prompts the question of whether a direct link exists between HOTAIR and miR-122 in promoting HCV-HCC, and if so, what would be the therapeutic implications of a combined therapy with HOTAIR antisense drugs? Similarly, although a direct link between lncRNA PVT1 and HCV-HCC is still lacking, PVT1 is known to be induced by HCV in HCC cells [67], and it has also been found significantly upregulated in HCC patients with worse prognosis [151,152]. The specific roles of PVT1 in cancer are still not clearly understood; however, it has been demonstrated to promote cell proliferation and the acquisition of stem cell-like properties in HCC cells by stabilizing NOP2 (nucleolar protein 2) [151] or, in other tumors, by downregulating the transcription of MYC (Myc proto-oncogene protein), its upstream neighboring gene [153]. Finally, lncRNA NORAD (non-coding RNA activated by DNA damage) has also been linked to the proliferation of HCV-infected HCC cells [154]. In this case, binding of NORAD to miR-373 releases their common target Wee1 (G2 checkpoint kinase), favoring hepatocyte growth due to Wee1-mediated cell cycle progression.

## 7. Conclusions

In summary, there is significant evidence of the profound effect that HCV-infection has on the transcription of human lncRNAs. However, given the abundance of lncRNAs in the human genome and its functional relevance, many more lncRNAs are expected to be deregulated after infection and playing relevant roles in HCV-infected cells. Most studies performed to date in cultured cells infected with HCV have reported lncRNAs that affect viral release or the antiviral response (Table 1). It is expected that other lncRNAs will be described in the future, affecting other steps of the viral cycle, including the replication of the virus, where several RNA binding proteins are involved that could act as lncRNA carriers. It is mandatory to determine whether lncRNAs that function to control the HCV cell cycle in cultured cells have a significant impact on progression from acute infection to chronicity; liver inflammation; and the development of fibrosis, cirrhosis, and HCC in infected patients. Given the high number of people infected with HCV worldwide, these studies may have a strong clinical impact. In fact, it is disappointing that only a few lncRNAs studied in HCV-infected cultured cells have also been evaluated in infected patients, and their clinical impact has not been addressed.

Further studies should also be performed in HCV-related liver diseases. To date, it is unclear whether HCV infection produces a unique HCV-related lncRNA signature in cirrhotic livers and HCCs that it is not found when these diseases result from other insults such as alcohol consumption, unhealthy eating, or infection with HBV. In fact, the immune response, the pro-fibrogenic and neoplastic transforming events that may take place after chronic HCV infection, generates a mixture of indirect events, unleashed in cascade in response to the virus, which should also occur in response to other chronic sources of liver damage. Therefore, it is highly probable that most of the deregulated lncRNA transcriptome is shared between HCV-related and -unrelated liver diseases. In the case of lncRNAs in HCV-HCC, where most studies have been performed thus far, it surprises the very few common findings between different studies (Table 2). This may reflect tumor heterogeneity, the complexity of the microenvironment, and the influence of host and environmental factors added to the variability observed at a cellular scale. Unsurprisingly, the sum of all these elements may have a high impact on lncRNA regulation and explain, at least in part, the little overlap that is found between different patient cohorts. Although highly complex, extensive transcriptomic studies should be performed in large cohorts of patients, in which the several etiologies that lead to HCC are well represented, the clinical data are well-annotated, and there is an excellent record of treatment and patient response. Such studies should be encouraged as they can open new avenues to explore for novel therapeutic targets as well as biomarkers that could aid in the diagnosis, prognosis, and patient management of liver-related diseases, especially in the context of HCV infection. 

## Figures and Tables

**Figure 1 ijms-21-02255-f001:**
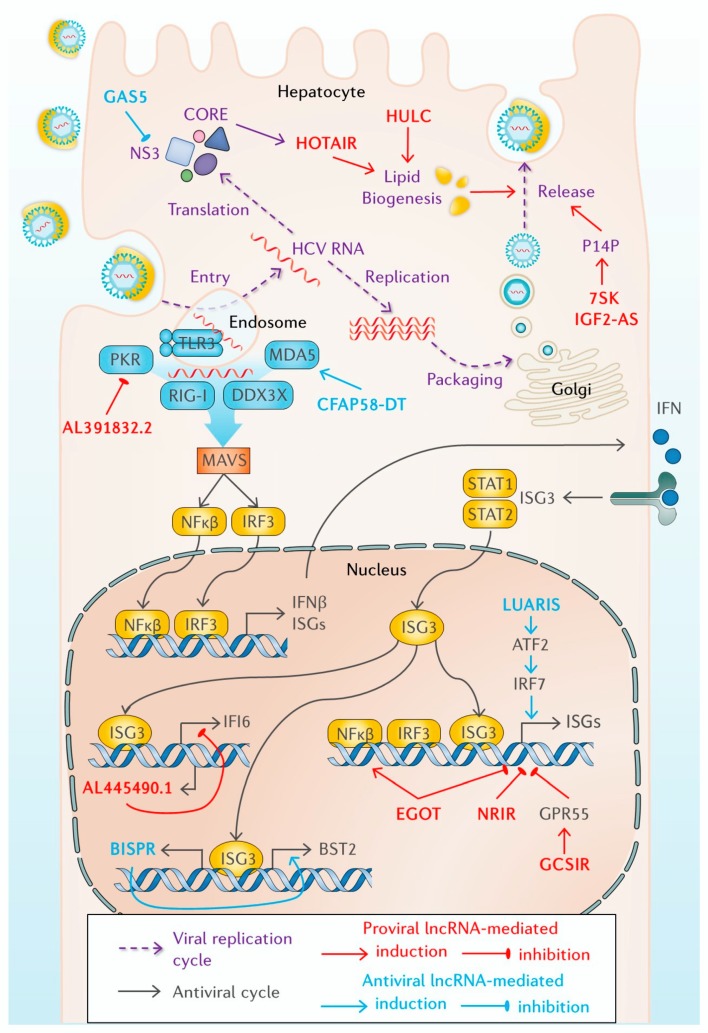
Long non-coding RNAs (lncRNAs) deregulated during the replication cycle or the antiviral response of hepatitis C virus (HCV)-infected cells. See text for details.

**Figure 2 ijms-21-02255-f002:**
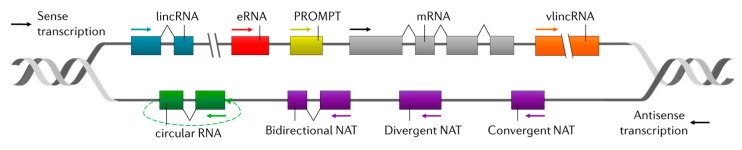
Classification of lncRNAs according to their genomic position. See text for details.

**Figure 3 ijms-21-02255-f003:**
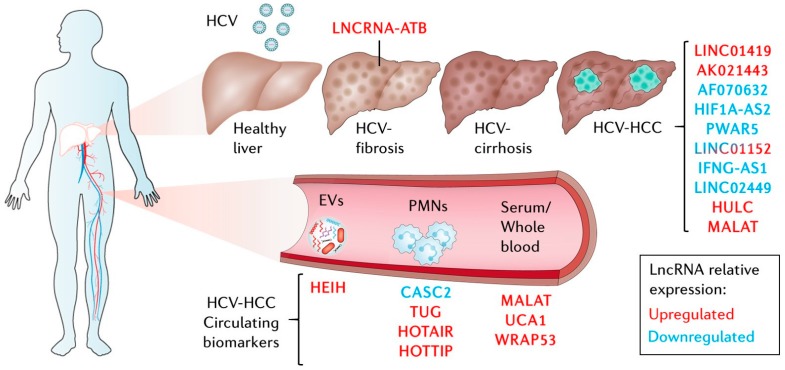
LncRNAs deregulated in HCV-hepatocellular carcinoma (HCC) in tumor samples and cell studies or as circulating biomarkers in liquid biopsies. EVs: extracellular vesicles, PMNs: polymorphonuclear leukocytes. See text for details.

**Table 1 ijms-21-02255-t001:** LncRNAs deregulated in HCV viral cycle and antiviral response.

GENCODE Name	Alternative Name	HCV	Regulation	Impact	Role	Function	References
GAS5	-	Up		5′end blocks NS3	Blocks NS3	Antiviral	[71]
HOTAIR	-	Up	Core	Silences SIRT1 promoter and affects glucose and lipid metabolism	Viral release	Proviral	[74]
HULC	-	Up	RXRA	Required for lipid biogenesis. Facilitates binding of core with lipid droplets	Viral release	Proviral	[75]
IGF2-AS	-	Up	STAT3	Increases phosphatidylinositol 4-phosphate (PI4P)	Viral release	Proviral	[76]
7SK	-	Up	STAT3	Increases phosphatidylinositol 4-phosphate (PI4P)	Viral release	Proviral	[76]
AL391832.2	lncATV	Up	IFN	Binds to RIG-I and inhibits signaling and IFN production	Antiviral response	Proviral	[77]
CFAP58-DT	lncITPRIP-1	Up	IFN	Binds MDA5 and induces oligomerization, activation, and binding to viral RNA	Antiviral response	Antiviral	[20]
AL445490.1	lncRNA-IFI6	Up		Inhibits expression of neighbor IFI6, by affecting histone modification	Antiviral response	Proviral	[78]
BISPR	-	Up	IFN	Induces BST2 expression	Antiviral response	Antiviral	[79,80]
LUARIS	lncRNA-32		pI:C/down IFN	Binds ATF2 and induces ISG expression	Antiviral response	Antiviral	[81]
NRIR	lnc-CMPK2	Up	IFN	Negative regulator of several ISGs	Antiviral response	Proviral	[82]
GCSIR	lnc-ITM2C-1	Up	HCV/down IFN	Promotes expression of neighbor GPR55, negative regulator of ISGs	Antiviral response	Proviral	[83]
EGOT	-	Up	IFN/NF-κB	Negative regulator of several ISGs	Antiviral response	Proviral	[67]

**Table 2 ijms-21-02255-t002:** LncRNAs deregulated in HCV-related HCC and their functions in HCC and cancer.

GENCODE Name	Alternative Name	Relative Expression	Function in HCC	References
LINC01419	PRLH1	up	Promotes cell proliferation by helping HR repair of DNA damage	[119,120]
AK021443		up	Promotes cell cycle progression ^1^	[119]
AF070632		down	Involved in cofactor binding and cell metabolism ^1^	[119]
HIF1A-AS2	aHIF	down	Unknown	[117]
PWAR5	PAR5	down	Unknown	[117]
IFNG-AS1	TMEVPG1	down	Activates IFN-γ production and the antiviral response	[117,122]
LINC01152		up/down	Promotes tumorigenesis by binding IL-23 and activating STAT3/pSTAT3 in HBV-related HCC	[117]
LINC02499	lnc-AFM-2	down	Overexpression inhibits HCC cells proliferation, migration, and invasion in vitro	[123,124]
HULC		up	Enhances tumorigenesis and metastasis via activation of miR-200a-3p/ZEB1 pathway	[125,126]
MALAT1		up	Acts as a proto-oncogene through Wnt activation and induction of SRSF1 splicing	[127,128]
			**Biomarkers in Liquid Biopsies**	
MALAT1		up ^2^	Same as above	-
UCA1		up ^2^	Contributes to HCC through inhibition of miR-216b and activation of FGFR1/ERK pathway	[62,129]
WRAP53		up ^2^	Promotes cancer cell survival by regulating p53	[62,130]
HEIH		up ^3^	Promotes tumor progression by binding to a causing the repression of EZH2 target genes	[131,132]
CASC2		down ^4^	Suppresses EMT of HCC cells through CASC2/miR-367/FBXW7 axis	[133,134]
TUG1		up ^4^	Promotes cell growth and apoptosis by epigenetically silencing of KLF2	[133,135]
HOTAIR		up ^5^	Promotes tumorigenesis by suppression of miR-122 and activation of Cyclin G1	[136,137]
HOTTIP		up ^5^	Unknown in HCC; promotes cell proliferation and invasion by modulating HOXA13 in pancreatic cancer	[136,138]

^1^ predicted, not validated, ^2^ in serum, ^3^ in serum and exosomes, ^4^ in leukocytes, ^5^ in whole blood.

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
