# Peer review of "LncRNAs in HCV Infection and HCV-Related Liver Disease"

_ijms, 2020, doi:10.3390/ijms21062255_

Round 1

Reviewer 1 Report

The review article titled "LncRNAs in HCV infection and HCV-related liver disease" was well written and adequately described the topic of interest.

I didn't feel any issues with this article. I suggest to accept the article in its current form.

Thank you.

Author Response

We would like to thank the reviewer for his/her work

Reviewer 2 Report

Major comments:

  1. LncRNA is the main content of this manuscript. Lines 113-122, this part described the another non-coding RNA, such as (rRNAs) and transfer RNAs (tRNAs). Thus, this part should be simplified appropriately.
  2. As a great review, figure 2 should be drawn by yourself.
  3. Author introduced the HCV lncRNAs. I'm curious about whether the HCV lncRNA and human lncRNA have homology or HCV lncRNA could be transfer to human.
  4. The part “3.1.3. LncRNAs that affect the antiviral response” is complicate. Should it be divided into two parts for elaboration?
  5. Lines 358-373, plesase merge the two paragraphs into one paragraph.

Minor comments:

  1. Figure should appear where first mentioned.
  2. Figure 2 is not mentioned in the manuscript.
  3. Line 167, HOTAIR or PVT1 should be showed the full name where first mentioned. Please check for similar situations throughout the manuscript.
  4. Lines 230-232, please add the references.
  5. I am not sure whether there is any requirement for references in the magazine. It is recommended to reduce the number of references.

Author Response

We thank the reviewer for his/her work in reviewing this article. We believe his/her comments have helped us to improve the manuscript.

Major comments:

  1. LncRNA is the main content of this manuscript. Lines 113-122, this part described the another non-coding RNA, such as (rRNAs) and transfer RNAs (tRNAs). Thus, this part should be simplified appropriately.

The reviewer is right that this part was not necessary for specialized readers. We have simplified the reference to other non-coding RNAs. However, we decided to keep two lines about miRNAs since they are mentioned several times later in the text as part of the mechanism of action of various lncRNAs.

  1. As a great review, figure 2 should be drawn by yourself.

We agree with the reviewer on the importance of drawing our own figures. In fact, figure 2 was drawn by ourselves. Initially, we kept a reference to the figure it was based on; however, we have now removed this reference as we believe the figures share little resemblance. We have made additional changes that we thought were more relevant for the purpose of this review. For comparison, we attach both figures.

Figure from Gesualdo et al., 2014. Oncotarget. 5 (22), 10976-96. (please, see the attachment)

New figure 2. (please, see the attachment)

  1. Author introduced the HCV lncRNAs. I'm curious about whether the HCV lncRNA and human lncRNA have homology or HCV lncRNA could be transfer to human.

We are also amazed by HCV lncRNAs. They are lncRNAs because they are degradation products of the viral genome that have lost the IRES, and cannot be translated. Therefore, most of their sequence is shared with the viral genome and lack any homology with any human coding or non-coding sequence. Similar to the reviewer, we thought that some viral RNA could be transferred to human. We screened the transcriptome of HCV-infected cells looking for fussion transcripts with human and viral sequences. Although we did not find signs of such fussion transcripts, this is a negative result. The depth of the analysis and the bioinformatics tools used does not allow us to be certain whether fussion transcripts exist. Finally, HCV lncRNAs and/or fussion transcripts, if they exist, may be stable byproducts of RNA processing with no function. The possibility that HCV lncRNAs have a relevant function in the replication cycle of the virus is appealing, but only a hypothesis nowadays. We belive that further work is required to address this issue before we can give more relevance to this part in the review.

  1. The part "3.1.3. LncRNAs that affect the antiviral response" is complicate. Should it be divided into two parts for elaboration?

The reviewer is right. We have now subdivided this section into two parts.

  1. Lines 358-373, plesase merge the two paragraphs into one paragraph.

The two paragraphs have been merged in the revised manuscript.

Minor comments:

  1. Figure should appear where first mentioned.

In the manuscript initially submitted to IJMS, figures were placed right after their first mention. However, for editorial reasons beyond our reach, this was modified in the version for reviewers.

  1. Figure 2 is not mentioned in the manuscript.

There is a reference to figure 2 (Fig. 2) now in line 144 of the manuscript.

  1. Line 167, HOTAIR or PVT1 should be showed the full name where first mentioned. Please check for similar situations throughout the manuscript.

We thank the reviewer for this observation. This has been changed in the revised version. Careful revision of similar situations throughout the paper has also been done and appropriate changes have been made.

  1. Lines 230-232, please add the references.

An adequate reference has been added.

  1. I am not sure whether there is any requirement for references in the magazine. It is recommended to reduce the number of references.

To our knowledge, there is no limitation for the number of citations in a review paper. As a reference, the last review published by IJMS that we found had a total of 164 references. Nonetheless, we have reduced the number of references where possible, and we now cite 159 articles.

Round 2

Reviewer 2 Report

This work brings out an important notion which can be considered crucial on the role of lncRNAs in HCV infection and HCV-related liver disease. However, there are some comments to improve the manuscript.

  1. Line 38, please correct the style of “[this special issue, [6]”.
  2. In my opinion, the section ” LncRNAs in HCV infection” is also too much and a bit confusing. I suggest to rearrange the content as the following directory. Of course, this is only a suggestion. The author can modify according to the actual situation.
    1. Hepatitis C Virus (HCV)
    2. Long non-coding RNAs (LncRNAs)
    3. HCV lncRNAs
    4. Cellular lncRNAs related to HCV cell cycle and the antiviral response

        4.1 LncRNAs that affect the viral cell cycle

        4.2  LncRNAs that affect the antiviral response.

          4.2.1.  IFN-related lncRNAs that affect HCV replication

          4.2.2.  LncRNAs affecting the IFN synthesis pathway

          4.2.3  LncRNAs affecting the expression of specific ISGs located nearby

         4.2.4 LncRNAs affecting general ISG transcription

    1. LncRNAs in HCV-related liver diseases.

        5.1 LncRNAs related to extrahepatic manifestations and HCV-related liver fibrosis

        5.2 LncRNAs related to HCV-related hepatocellular carcinoma

        5.3 LncRNAs in liquid biopsies as biomarkers in HCV-HCC

        5.4 LncRNAs with potential HCV-HCC roles from cell studies

     6. Conclusions

Author Response

We want to thank the reviewer for helping us to improve the manuscript.

1.  Line 38, please correct the style of “[this special issue, [6]”.

Corrected

2. In my opinion, the section ” LncRNAs in HCV infection” is also too much and a bit confusing. I suggest to rearrange the content as the following directory. Of course, this is only a suggestion. The author can modify according to the actual situation.

We have arrange the order of the review as suggested by the reviewer with the few exceptions highlighted below

  1. Hepatitis C Virus (HCV)
  2. Long non-coding RNAs (LncRNAs)
  3. General considerations of HCV and lncRNAs
  4. HCV lncRNAs
  5. Cellular lncRNAs related to HCV cell cycle and the antiviral response.

    5.1 LncRNAs that affect the viral cell cycle

    5.2. LncRNAs that affect the antiviral response.

           5.2.1. IFN-related lncRNAs

           5.2.2.  IFN-related lncRNAs that affect HCV replication

            5.2.3.  LncRNAs affecting the IFN synthesis pathway

            5.2.4  LncRNAs affecting the expression of specific ISGs located nearby

             5.2.5 LncRNAs affecting general ISG transcription

  1. LncRNAs in HCV-related liver diseases.

             6.1 LncRNAs related to extrahepatic manifestations and HCV-related liver fibrosis

             6.2 LncRNAs related to HCV-related hepatocellular carcinoma

             6.3 LncRNAs in liquid biopsies as biomarkers in HCV-HCC

             6.4 LncRNAs with potential HCV-HCC roles from cell studies

  1. Conclusions